# Perceived Needs, Barriers, and Challenges to Continuing Professional Development (CPD): A Qualitative Exploration among Hospital Pharmacists

**DOI:** 10.3390/pharmacy12050140

**Published:** 2024-09-12

**Authors:** Sundus Aldakhil, Sana Majdi Baqar, Bashayr Alosaimi, Rafal Almuzirie, Maryam Farooqui, Saud Alsahali, Yasser Almogbel

**Affiliations:** Department of Pharmacy Practice, College of Pharmacy, Qassim University, Buraidah 51452, Qassim, Saudi Arabia; sundusaldakhil@gmail.com (S.A.); sanabaqarx@gmail.com (S.M.B.); bshayeraalosaimi@gmail.com (B.A.); revalmohammmed@gmail.com (R.A.); s.alsahali@qu.edu.sa (S.A.); y.almogbel@qu.edu.sa (Y.A.)

**Keywords:** continuing professional development, practice guidelines, education, hospital pharmacists, qualitative study, Saudi Arabia

## Abstract

Continuing professional development (CPD) is an essential tool for healthcare professionals to remain up-to-date with the latest advancements in their field. In Saudi Arabia, pharmacists are officially registered healthcare professionals by the Saudi Commission for Health Specialties (SCFHS). To uphold their licensure, they must complete 40 h of CPD every two years. This qualitative study aimed to explore hospital pharmacists’ perceptions, barriers, and challenges of CPD, as well as their recommendations for improving CPD activities. A qualitative descriptive approach with semi-structured face-to-face interviews was employed. Using purposive sampling, 12 hospital pharmacists were interviewed. The recorded data were transcribed and analyzed using thematic analysis. Pharmacists generally showed positive perceptions of CPD, recognizing its importance for their professional development and the provision of high-quality patient care. However, some pharmacists expressed dissatisfaction with the current system. Several barriers to participation such as heavy workloads, lack of time, and limited financial support were highlighted. The primary motivations for engaging in CPD included fulfilling regulatory requirements. Workshops were the most beneficial CPD activities among others. Additionally, importance of more specialized, practice-oriented programs was highlighted. The study provides valuable insights into the needs and challenges faced by hospital pharmacists in Saudi Arabia regarding CPD. The insights gained can inform theory, policy, and practice relating to pharmacists’ CPD at both professional and governmental levels.

## 1. Introduction

Being a healthcare professional necessitates staying current with the most recent advancements in the field [1]. Due to the rapid advancement of science and the paramount concern for patient safety, pharmacists have a duty to remain up-to-date on the most recent developments and evidence in pharmacy practice, pharmaceutical sciences, and related areas within their profession [2].

This sheds light on the importance of continuing professional development (CPD). CPD has been defined by the International Pharmaceutical Federation (FIP) as “the responsibility of individual pharmacists for systemic maintenance, development and broadening of knowledge, skills and attitudes, to ensure continuing competence as a professional, throughout their careers” [3]. It represents pharmacists’ ongoing, self-directed dedication to guaranteeing the safe and efficient provision of pharmaceutical care to patients [4].

Furthermore, the FIP global pharmacy education vision and standards, published at the end of 2016, address CPD as one of the essential abilities that modern pharmacy curricula should emphasize to guarantee that graduates have the skills and abilities needed to keep learning throughout their careers [5]. In 2016, it was evident that one of the key issues faced in Saudi Arabia was the ongoing demand for highly skilled pharmacists. These professionals are required to run core clinical pharmacy services and provide training sites for students and residents across various significant fields [6]. According to the latest statistics by the ministry of health, the Saudi pharmacy workforce has nearly doubled in size over the past five years, reaching a peak of 12,058 pharmacists in 2021. Hospital pharmacies form the second largest sector with 9121 pharmacists making up (29.6%) of the total workforce [7]. Therefore, efforts are being made to enhance pharmacy education in Saudi Arabia to ensure that it matches international standards.

To improve pharmacists’ professional abilities, each country has developed its own way of gaining CPD hours [8]. Participating in such an activity is mandatory to earn professional learning credits annually to meet performance standards. The professional development of pharmacist staff is critical to the healthcare system’s quality of care [1].

Today, many countries like the United States and Australia have applied the mandatory set number of credits system for pharmacist license renewal [9,10]. In Great Britain, revalidation is an annual requirement for every pharmacy professional to renew their registration and continue practicing. As part of revalidation, four CPD records, one peer discussion record, and one reflective account record are required [11,12]. Pharmacists in Jordan are required to complete 50 h of CPD annually to renew their licenses [13]. In Arabian Gulf Countries, Saudi Arabia, Qatar, and the United Arab Emirates, the CPD systems are used [14,15,16]. In Saudi Arabia, the Saudi Commission of Health Specialties (SCFHS) oversees CPD activities which includes health and medical training courses, conferences, seminars, or lectures designed to advance health practitioners’ skills and to obtain CME (Continuous Medical Education) hours [17]. More crucially, the Ministry of Health (MOH) mandates 40 CPD hours by pharmacists to be completed every two years [18]. Different activities can be attended to accomplish the required hours. Attending conferences, seminars, workshops, training sessions or research, journal articles, and book publications can offer up to 25 h [18,19].

Although a report published in 2021 showed that Saudi pharmacists appeared to have a great craving for CPD [20], a previous study on identifying pharmacy workforce needs in Saudi Arabia mentioned that despite the availability of CPD hours and that a certain number of hours are mandatory for re-registration, these hours do not address the real need of a pharmacist for professional development. Moreover, CPD hours may not be related to a pharmacist’s specialty [19]. Expert panel members reached the consensus that the current CPD system needs to be shifted to individualized and comprehensive CPD to focus on the actual needs of pharmacists [21].

Exploration of pharmacists’ perspectives on CPD, including challenges and barriers, is crucial, as this information can help providers of such programs and policymakers address pharmacists’ expectations and challenges and develop content that is both relevant and motivating, which will ultimately improve the learning experience and patient care [22]. Therefore, this study aims to explore challenges and barriers faced by hospital pharmacists when they are seeking education and development in the pharmacy field. Hospital pharmacists play a critical role in the healthcare system as they are directly involved in patient care, medication management, and interdisciplinary collaboration within hospital settings. Their specialized knowledge and clinical expertise make them key contributors to optimizing treatment outcomes and enhancing overall health care quality. Furthermore, this study investigates future recommendations and suggestions to improve CPD activities among hospital pharmacists.

## 2. Materials and Methods

### 2.1. Study Design and Sampling

For an in-depth exploration of pharmacists’ perspectives, this study was conducted using a qualitative descriptive approach with semi-structured face-to-face interviews. Purposive sampling was used. To ensure the quality of the research, the consolidated criteria for reporting qualitative research (COREQ) checklist was used [23].

### 2.2. Study Location

This study was conducted among hospital pharmacists from four major hospitals (King Saud Medical City (KSMC), Riyadh; Al-Shifa Hospital, Qassim; King Saud Hospital, Qassim; and Maternity and Children Hospital, Qassim) located in Qassim and Riyadh provinces. KSMC is a tertiary hospital with more than 1400 beds to serve the local population in the Riyadh region, which is the capital city of Saudi Arabia. Also, King Saud Hospital is a secondary general hospital which has many specialties with about 300 beds to serve the local population in Unaizah city, Qassim region. On other hand, the Maternity and Children Hospital and Al-Shifa Hospital are specialized hospitals offering services in pediatric and geriatric care, respectively, which located in the Qassim region of Saudi Arabia. By including hospitals from both the Riyadh and Qassim regions, this study aimed to capture a diverse perspective on the factors influencing pharmacists’ participation in CPD activities across different regions of Saudi Arabia. This approach allows for a more comprehensive understanding of the contextual factors that may influence pharmacists’ engagement with CPD in these specific regions.

### 2.3. Sample and Sampling Method

A purposive sampling approach was employed to recruit pharmacists for this study. Eligible criteria for participation included holding a pharmacy qualification certificate and having a minimum of six months of experience working in a hospital setting. Pharmacists without relevant clinical experience were not included in the study. Additionally, interns and trainees were excluded from participation.

### 2.4. Study Tool

Data for this study were collected using face-to-face semi-structured interviews. The interview guide (Table 1) was developed based on a comprehensive literature review [24,25,26,27,28]. By conducting face-to-face interviews, we were able to establish a rapport with the participants, create a comfortable environment for sharing their experiences, and probe further into their responses to gain deeper insights.

### 2.5. Data Collection

After obtaining the ethical approval, the head of the pharmacy division of each hospital was approached to indicate potential participants. Eligible pharmacists were approached and provided with a clear overview of the study objectives. The pharmacists were given the opportunity to express their interest in participating. For those who expressed interest, suitable interview time and dates were organized in coordination with each participant to ensure their convenience and availability. The participants were then asked to provide informed consent by signing a consent form indicating their willingness to take part in the study. Once informed consent was obtained, face-to-face semi-structured interviews were conducted. The interviews were audio-recorded with the participants’ consent to ensure accuracy and capture the richness of their responses. This approach ensured transparency and adherence to ethical guidelines throughout the data collection process while respecting the autonomy and informed participation of the pharmacists involved.

### 2.6. Data Analysis

The recorded interviews were transcribed verbatim to capture the participants’ responses in their exact words. These transcripts underwent thorough review and analysis by two different authors to ensure accuracy and a comprehensive understanding of the data. Following Braun and Clarke, a systematic method for thematic analysis (TA) was employed to examine the data [29]. TA allowed for the identification, organization, and insight into patterns of meaning, known as themes, that emerged across the dataset [29]. Through the process of TA, different themes were extracted, representing recurring patterns and significant findings related to the factors influencing pharmacists’ participation in CPD activities. Interviews were stopped after reaching the thematic saturation, where no new themes emerged from the subsequent interviews.

### 2.7. Rigor

Lincoln and Guba (1985) proposed four criteria to assess the overall trustworthiness of qualitative research results. These criteria serve as benchmarks for evaluating the validity and reliability of a qualitative study. The four criteria are as follows: credibility, transferability, confirmability, and dependability [30]. To ensure and impart credibility, the interviews were guided by interview questions based on a comprehensive review of literature. We invited seven clinical pharmacy experts outside the team to review it. To ensure that the guide was understandable and relevant, the guide was piloted prior to the execution of the study with an academic pharmacist who has background experience in hospital settings. Transferability was ensured by providing detailed methodology including study design, study tool, sample, sampling method, and data analysis [29]. For the confirmability and dependability, it was enhanced through recording and interpreting the verbal and non-verbal data. Additionally, two researchers independently analyzed the data and double-checked the translation of the interviews [31].

### 2.8. Ethical Consideration

The study was approved by the Ethics Committee at Qassim Region (reference number 607-45-11498). The participants were informed about the study and gave written informed consent, emphasizing the voluntary nature of participation and the option to withdraw at any point. Additionally, the participants were assured that their identities would remain confidential throughout the reporting process. Access to the data was restricted to the researchers and the research team.

## 3. Results

A total of 12 pharmacists participated in the interviews. The characteristics of the pharmacists are presented in Table 2. Through the process of data analysis, six major themes were identified and confirmed Table 3: understanding and importance of CPD, CPD hours for renewal and other motivating factors, barriers and obstacles to CPD participation, preferences regarding CPD activities, the impact of COVID-19 on the rapid development of CPD, and future recommendations. Each major theme further contained different subthemes, capturing a comprehensive range of perspectives and insights from the participants.

### 3.1. Understanding and Importance of CPD

All the interviewed pharmacists showed that the concept of CPD is clear. One pharmacist defined CPD as follows: *“CPD stands for Continuing Professional Development, which is similar to CME (Continuing Medical Education). It encompasses various activities such as lectures, courses, and workshops aimed at providing professionals with ongoing learning opportunities to maintain and enhance their skills and knowledge in their respective fields”.—Pharmacist 11*.

Another definition provided by a different pharmacist was, *“It is about post-graduation training, courses, lectures, and conferences designed to keep healthcare practitioners up to date with the latest developments in their field. The goal is to ensure that they practice modern healthcare according to global standards”.—Pharmacist 12*.

The interviewed pharmacists perceived CPD as crucial for their career growth and the quality of patient care. They viewed CPD as a means to maintain and enhance their professional skills, knowledge, and job performance. Engaging in CPD activities allowed them to stay up-to-date with the latest medical knowledge, treatment guidelines, and pharmacy practices, which are essential for providing safe and effective patient care. They also believed that CPD supports them in adapting to new roles and technologies in the healthcare sector, ultimately enhancing the quality of healthcare services. Overall, the interviewed pharmacists showed their satisfaction with the current CPD system.

However, some pharmacists stated that they were not fully satisfied with the current system. First, they thought that the current CPD program could not be considered true professional development, stating, *“I think what we have is continuing education, not development”.* They argued that the program lacked indicators or evaluations to assess pharmacist development. Second, pharmacists felt that the system lacked motivation beyond simply accumulating hours. They asserted that the focus was on collecting hours without considering the quality or relevance of the information provided. One pharmacist explained, *“They ask for hours, not for the topic. This is the issue”.* Third, when comparing the available local programs with international ones, pharmacists believed that they were lagging behind. One pharmacist shared their experience, saying, *“I had a question for the last 11 years, and I only received the answer three months ago by attending sessions led by experienced American professionals in IV and PN through my membership with ASPEN and ASHP. It has been incredibly valuable. In just three lectures, I gained knowledge equivalent to my 18 years of experience”.—Pharmacist 9*.

### 3.2. CPD Hours for Renewal of Pharmacy License and Other Motivations

Motivation to engage in CPD activities can be categorized into two main factors: regulatory requirements and personal interest. For pharmacists, fulfilling the required number of CPD hours linked to license renewal remains a primary motivation. Additionally, meeting CPD requirements is now connected to annual performance assessments, professional performance certifications, and overall professional valuation. These developments have contributed to the increasing significance of CPD hours as a mandatory component for pharmacists to maintain their professional standing and ensure their competence in their respective fields.


*“Most of us attend the course because of the hours; we need the hours for health certificate renewal”.—Pharmacist 4*



*“The perception of CPD among pharmacists primarily revolves around fulfilling the necessary hours for license renewal. Many pharmacists view CPD as a requirement to maintain their qualifications, rather than as an opportunity for professional growth or skill enhancement”.—Pharmacist 11.*



*“Besides the mandatory hours, we also need these hours for the yearly employee evaluation in our department”.—Pharmacist 3*


On the other hand, many pharmacists are motivated to participate in CPD activities not only due to regulatory requirements but also because of personal interest in specific topics. They seek out relevant CPD opportunities to enhance their knowledge and skills. Self-encouragement, self-education, and a desire for professional growth drive them to actively engage in CPD. Additionally, a supportive job environment and encouragement from managers play a role in motivating pharmacists to pursue CPD opportunities.

### 3.3. Barriers and Obstacles

Pharmacists face several challenges and barriers that hinder their participation in CPD. These challenges include:

#### 3.3.1. Time

Many CPD activities are scheduled during work hours, making it difficult for pharmacists to attend. Even when activities are held on weekends, personal commitments can pose challenges to participation.


*“The time barrier arises because there are often excellent courses that we cannot attend due to our professional commitments”.—Pharmacist 12*



*“It is the time. As an employee, I want to attend a workshop in Riyadh, but I cannot get a day off. Additionally, most activities are scheduled on weekdays, making it difficult for us”.—Pharmacist 1*



*“Honestly, it is the time. Many activities take place during work hours, making it hard for us to attend. Perhaps they should consider scheduling them on weekends, but even then, weekends are our free time, and we have other commitments”.—Pharmacist 2*


#### 3.3.2. Limited Program Specialization in Pharmacy

Many pharmacists found that the current CPD programs available were not specialized enough for the pharmacy profession. They struggled to find activities directly relevant to their practice and emphasized the need for more comprehensive coverage in the field.


*“They focus on medicine and nursing, neglecting other specialties”.—Pharmacist 4*



*“I believe we should tailor CPD programs to specific professions like pharmacy. It is crucial to address the actual needs of practitioners”.—Pharmacist 11*


#### 3.3.3. Resistance to Change and Shifting Priorities

Older pharmacists mentioned that as they age and take on more responsibilities, their priorities may change. The new generation shows more willingness to engage in continuous development, while some older pharmacists tend to stick to the principles they adopted earlier in their careers.


*“As you get older, you become less active; that is the main challenge. When you start a family, your priorities change, and your mind is occupied with various things”.—Pharmacist 4*



*“I think the new generation is more eager to participate in CPD activities. Most current employees are young, and the resistance comes from the older generation”.—Pharmacist 2*


#### 3.3.4. Costs

Attending CPD activities can pose a financial burden, especially when travel or registration fees are involved. Some pharmacists perceive a lack of quality in free courses and see a shortage of specific topics, which allows program providers to take advantage of pharmacists’ needs and raise costs.


*“It was very expensive, and some people exploit the demand by raising prices”.—Pharmacist 1*



*“The issue is money. Some fees are very costly. For online courses, aside from the fee, there are usually no barriers, but for other activities, it depends on my working hours and whether I can take time off”.—Pharmacist 3*


#### 3.3.5. Lack of Employee Availability or Support

Some pharmacists mentioned that despite the availability of CPD activities, coordinating with other employees and obtaining days off can hinder their participation.


*“The main challenge is the lack of support and coordination with other employees, making it difficult to attend CPD activities”.—Pharmacist 4*


### 3.4. Preferences

Pharmacists often have specific preferences that guide their choice of learning formats. These preferences are influenced by their goals, desire for interactive learning experiences, and geographical constraints. Understanding these preferences is essential for CPD providers to design effective programs that cater to pharmacists’ needs.

#### 3.4.1. Goal-Oriented Approach

Pharmacists approach CPD with specific goals in mind that influence their choice of learning format. When the primary goal is to fulfill CPD hours, online courses are often preferred due to their flexibility and convenience.


*“I recognize the value of workshops and conferences, particularly when practical skills are involved, such as in IV training. When my primary goal is to fulfill CPD hours, online courses are my go-to option”.—Pharmacist 11*



*“In terms of hours, we prefer online sessions for their flexibility. Hence, for genuine development, we advocate for a holistic approach”.—Pharmacist 5*


#### 3.4.2. Interactive Learning

Workshops excel at promoting interaction and communication among participants. Unlike online sessions or conferences, workshops offer intimate group settings that encourage active participation and facilitate rich discussions. This direct engagement enhances learning outcomes and fosters a supportive environment conducive to professional growth.


*“The opportunity for active participation and focused discussions within a smaller group enhanced the learning experience and allowed for deeper exploration of complex topics”.—Pharmacist 1*



*“Workshops provide richer information and foster better communication compared to online platforms”.—Pharmacist 3*



*“Workshops stand out as my preferred mode… facilitate more engaging discussions with presenters and offers greater flexibility in learning”.—Pharmacist 7*


#### 3.4.3. Geographical Constraints and Accessibility

Geographical constraints also play a significant role in the choice between online learning and other formats. Many pharmacists prefer online CPD opportunities because of their accessibility, especially when workshops are not readily available in their regions.


*“We lean towards online CPD opportunities due to their accessibility. Many workshops are not held in the Qassim region, necessitating travel and time off from work, particularly for longer workshops spanning one to two days”.—Pharmacist 6*



*“Currently, there are many important and interesting activities that pharmacists would like to attend, but they are often held in one centralized location that may be far away for many professionals”.—Pharmacist 2*


### 3.5. The Impact of COVID-19 on the Rapid Development of CPD

The COVID-19 pandemic has led to a revolution in the use of online platforms [32]. During this period, there was increased accessibility to sessions and online learning platforms [33]. Pharmacists highlighted the shift from predominantly on-site sessions to online formats as one of the advantages of this pandemic, citing convenience and ease of access as major benefits. Before the pandemic, attending workshops often necessitated travel, disrupting work schedules and requiring time off.


*“For the last few years, after COVID-19, it is developed rapidly. Before, we could not even reach the 40 h that was mandatory by the Ministry of Health”.—Pharmacist 6*


However, with the transition to online platforms, individuals can now participate in CPD sessions from anywhere, eliminating the need for travel and accommodating busy schedules. Additionally, the pandemic accelerated the pace of CPD development, allowing the flexibility to attend multiple sessions per week and enabling pharmacists to accumulate CPD hours more efficiently. This shift emphasizes the transformative impact of COVID-19 on professional development practices and education, paving the way for more accessible and flexible learning opportunities.

### 3.6. Future Recommendations

While some pharmacists believe that the current CPD requirements adequately address the evolving needs of the pharmacy profession, others feel that there is a need to review and possibly update these requirements to ensure ongoing relevance. Suggestions for improvement include conducting research, taking feedback from professionals, and incorporating modern ideas, such as artificial intelligence (AI) and robotics, into CPD programs to enhance their effectiveness and relevance. Additionally, addressing the challenge of fragmented CPD platforms is crucial.

#### 3.6.1. AI-Powered Platforms


*“CPD is poised to undergo significant transformation in the foreseeable future, propelled by modern concepts such as AI and robotics. It is imperative to ensure a continuous update of this concept to remain aligned with these advancements”.—Pharmacist 12*


The integration of AI into CPD programs has immense potential to revolutionize the learning experience through the application of advanced technologies. By leveraging AI, CPD programs can personalize content, deliver real-time feedback, and provide intelligent recommendations for professional development. AI-powered platforms have the capability to analyze pharmacists’ individual learning patterns and identify specific knowledge gaps, enabling the suggestion of tailored learning modules to address their unique needs. Embracing AI in CPD programs can lead to enhanced adaptability, efficiency, and engagement, ultimately empowering pharmacists to remain at the forefront of their field. The incorporation of AI into CPD programs represents a significant opportunity for the optimization of continuous professional development in the pharmacy profession.

#### 3.6.2. Fragmented CPD Platforms


*“One of the main barriers I encounter when attempting to participate in CPD is the lack of a unified platform for CPD courses. Instead, there are numerous platforms available, each requiring separate payments. This fragmented system makes it challenging to keep track of courses and participate consistently”.—Pharmacist 11*



*“A website offering specialized courses for pharmacists should cater to hospital pharmacists, community pharmacists, and pharmacists in general. This will enhance pharmaceutical care quality across all practice settings”.—Pharmacist 10*


Having a unified platform for pharmacists to discuss and plan CPD activities throughout the year would be beneficial. This would streamline the process, making it easier for pharmacists to access courses, track their progress, and engage in continuous learning. Furthermore, a unified platform could offer personalized recommendations and curated content based on pharmacists’ interests and areas of expertise, enhancing the relevance and effectiveness of CPD activities. Ultimately, implementing a unified platform for CPD would not only streamline administrative tasks but also promote a culture of continuous learning and improvement within the pharmacy profession. Other suggestions include providing a unique and valuable program with a professional certificate to further enhance the effectiveness and value of CPD programs.

In light of these valuable suggestions, pharmacists passionately emphasized the need for greater emphasis on specific topics, including medication safety, quality, health leadership, chemotherapy, IV, and biological medication. However, their concerns extended beyond the mere availability of these topics. They underlined the importance of addressing the quality of content and the necessity of up-to-date materials.

## 4. Discussion

This pioneering study marks the first qualitative exploration of hospital pharmacists’ perceptions of CPD in Saudi Arabia. The findings of this study revealed a generally positive perception among pharmacists of CPD. The participants demonstrated a clear understanding of the importance of CPD for professional growth and development. However, there were varying opinions regarding satisfaction with the current CPD system. While some pharmacists believed that the existing programs adequately met their practice needs, others emphasized the need for tailored CPD activities specifically designed for the pharmacy profession. It is essential to address the actual needs of pharmacy practitioners when designing CPD programs. Aljadeed et al. (2024) conducted a study emphasizing the significance of aligning CE activities with the preferences and practices of pharmacy professionals [34]. This finding resonates with similar observations made in the nursing profession [35], indicating the importance of creating educational programs that cater to the specific needs of various professionals working in primary healthcare settings. This highlights the need to move beyond a focus solely on medical fields and to consider the professional requirements of other healthcare disciplines.

We found that the key motivations for engaging in CPD included meeting regulatory requirements, such as license renewal and professional assessments, as well as personal interest in specific topics and a desire for professional growth. Supportive work environments and encouragement from managers were also noted as contributing factors. Our finding agrees with Kandasamy et al.’s previous study which demonstrated that 78% of hospital pharmacists attended CPD activities to maintain their registration at SCFHS, and around the same number reported that the activities were relevant to their practice and were based on personal development needs [35].

In our study, several barriers and obstacles to CPD participation were identified. Time constraints emerged as a major challenge, with difficulties in attending activities scheduled during work hours or those that conflicted with personal commitments. Pharmacists reported challenges in finding the time to engage in CPD activities due to heavy workloads and competing priorities. Similar to the barriers reported in the current study, researchers in different countries have also identified time constraints and lack of motivation as factors that discourage pharmacists from participating in CE activities [15,36,37,38]. Saade et al. found that job restrictions associated with CE, as well as a lack of personal time and motivation, were the major barriers reported by a sample of 525 pharmacists [24]. Other key barriers to CPD participation included a lack of specialized or relevant program offerings, resistance to change among older pharmacists, and financial costs associated with attending CPD activities. This aligns with previous research by Kandasamy et al. in 2023, which identified cost as a major obstacle [36]. The existing literature has also consistently highlighted other significant barriers, including a general lack of resources available to pharmacists, restrictive CPD requirements, and perceptions among some professionals that they have already attained sufficient expertise to forgo further development [39,40].

Our results also suggest that pharmacists’ preferences for specific learning formats vary based on their goals and constraints, highlighting the importance of tailoring CPD programs to meet their needs. Online courses are favored by pharmacists for fulfilling CPD hours due to their flexibility, allowing professionals to learn at their own pace and convenience. The present study found that virtual/online attendance was preferred over live on-site CPD activities. This finding supports the results obtained by another study conducted in Canada after the COVID-19 pandemic [41]. The COVID-19 pandemic has had a significant impact on the rapid development of CPD, particularly in terms of increased accessibility to sessions and online learning platforms [42]. The shift to online platforms has provided pharmacists with more opportunities to engage in CPD activities from the comfort of their own locations. On the other hand, workshops are highly valued for their interactive nature, facilitating active participation and meaningful discussions among participants. Research from various countries has found similar benefits associated with this interactive approach [43,44,45,46].

Our research findings indicate a pressing need for the establishment of a unified platform for CPD programs in Saudi Arabia. Currently, there are several challenges faced by pharmacists in accessing and participating in CPD activities. These challenges include limited program specialization, fragmented resources, difficulties in tracking and managing CPD requirements, and a lack of standardized evaluation and assessment methods. By implementing a unified platform, these challenges can be effectively addressed.

### Limitations and Areas for Future Research

This study was limited to pharmacists working in a few hospitals in Saudi Arabia, and the findings cannot be generalizable to other healthcare settings outside Saudi Arabia due to variations in healthcare settings, cultural norms, and regulatory frameworks. Furthermore, we used a qualitative design, which limits the generalization of findings to the wider population. In addition, our data relied on face-to-face interviews, introducing potential interviewer and social desirability biases that limit verifying the accuracy of the data. However, replication of this research in different countries may highlight the influence of contextual and cultural factors on pharmacists’ engagement with CPD. In future, a more rigorous mixed-methods approach that combines self-reported surveys and interviews to explore pharmacists’ views and preferences for CPD courses can be conducted to explore the phenomena. While professional organizations undoubtedly offer valuable educational resources and networking opportunities through their CPD programs, future studies hold the potential for conducting more in-depth evaluations to scrutinize the impact of these activities on pharmacists’ professional development.

## 5. Conclusions

The current study provided an in-depth exploration of CPD among hospital pharmacists. Although the sample size was limited, the findings highlight several key recommendations to improve the CPD experience for pharmacists, as well as the challenges they face. One of the distinctive barriers identified was the lack of a unified platform for CPD courses. There is a clear need for a centralized platform that allows pharmacists to easily access, discuss, and plan CPD activities throughout the year while also overcoming cost constraints. Furthermore, there is a need for greater collaboration from employers to allocate working hours for pharmacists to attend CPD activities. Additionally, expanding the range of CPD topics and activities within the profession will be crucial to better fulfilling the diverse educational needs of pharmacists. The insights gained through this study provide a valuable foundation for policymakers, regulators, pharmacy organizations, and pharmacists to work toward enhancing the CPD landscape. By implementing evidence-based strategies, the quality and accessibility of CE for hospital pharmacists can be significantly improved, ultimately benefiting patient outcomes and the overall advancement of the profession.

## Figures and Tables

**Table 1 pharmacy-12-00140-t001:** Interview Guide.

Section	Questions
Understanding and experiences of CPD	What do you understand with the term continuing professional development (CPD)? When was the last time you attended a CPD activity, and what was the topic? Whose recommendation did you attend (your own, your boss’s, or a colleague’s)?, was that free or you had to pay? Do you prefer paid or free-of-charge CPD?
Preference in CPD activities	4. How do you prefer to engage in CPD (e.g., workshops, online courses, conferences), and which of these types do you find most beneficial to your practice?
Perceptions of CPD	5. How do you think the perception of CPD among pharmacists has evolved over the years? 6. How important do you perceive continuing professional development (CPD) to be in your career as a pharmacist? 7. Have you experienced any positive outcomes or advancements in your career as a result of participating in CPD?
Motivation towards CPD	8. What influences and motivates you to engage in CPD activities? 9. What barriers do you encounter when attempting to participate in CPD?
Future recommendations and suggestions to improve CPD activities among clinical pharmacists	10. Do you think the current CPD requirements adequately address the evolving needs of the pharmacy profession? 11. Are there any specific areas or topics within pharmacy practice that you feel require more emphasis on in CPD programs? 12. What improvements would you suggest to enhance the effectiveness and relevance of CPD programs?

**Table 2 pharmacy-12-00140-t002:** Characteristics of the Pharmacists.

No.	Gender	Age	Year of Experience	Education Level
1	Female	27	2	Bachelor
2	Male	30	3	Bachelor
3	Male	34	7	Bachelor
4	Male	40	17	Bachelor
5	Female	28	4	Clinical pharmacist PGY1
6	Female	32	6	Bachelor
7	Male	34	10	Bachelor
8	Male	37	14	Bachelor
9	Male	44	18	Bachelor
10	Male	40	17	Master
11	Male	31	5	Bachelor
12	Male	43	11	Master

**Table 3 pharmacy-12-00140-t003:** Major Themes and Subthemes.

Major Theme	Subtheme
Understanding and Importance of CPD	-
CPD Hours for Renewal of pharmacy license and Other Motivations	Regulatory Requirements Personal Interest
Barriers and Obstacles	Time Limited Program Specialization in Pharmacy Resistance to Change and Shifting Priorities Costs Lack of Employee Availability or Support
Preferences	Goal-Oriented Approach Interactive Learning Geographical Constraints and Accessibility
The Impact of COVID-19 on the Rapid Development of CPD	-
Future Recommendations	AI-Powered Platforms Fragmented CPD Platforms

## Data Availability

The datasets used for this study were available from the corresponding author upon reasonable request.

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
