# Peer review of "Perceived Needs, Barriers, and Challenges to Continuing Professional Development (CPD): A Qualitative Exploration among Hospital Pharmacists"

_pharmacy, 2024, doi:10.3390/pharmacy12050140_

Round 1
Reviewer 1 Report
Comments and Suggestions for Authors
Brief summary
This paper aims to and provides some insights into how hospital pharmacists in SA view CPD and suggests some possible solutions to issues. A straightforward qualitative descriptive methodology was employed with the use of semi-structured interviews. Similar insights/results to previous studies were found. Some suggestions as to how the paper could be improved in relation to context and methodology details are outlined in more detail below.
General concept comments
The terms CE and CPD are sometimes used interchangeably in the literature. The FIP definition of CPD is included in the introduction which is helpful. CE systems typically collect points or hours of learning. CPD systems employed by pharmacy regulators have moved to more goal orientated learning with the use of reflective cycles. Other countries (e.g. Great Britain) have moved further again and introduced revalidations systems which incorporate peer discussions/reflections alongside CPD cycles. The CPD system in SA was mentioned but not described in terms of where it sits within this context. Does the pharmacy regulator in SA simply expect the pharmacists to submit certificates to provide evidence of hours at the end of each year? If this context was given in the introduction it would lead to more meaningful suggestions around the platform suggested in the discussion and conclusion. Would this platform require more goal orientated learning with the use of reflection and application of learning?
What postgraduate support is in place for qualified pharmacists? Is there a deanery/postgraduate provider or are universities offering courses? Or is it all CME which would normally be geared to medics?
Specific comments
Lines 59 and 60.
Today, many countries like the United States and British, have applied the mandatory set number of credits system for pharmacist license renewal [9,10]. Mandatory CPD is also used in the United Kingdom and Australia [11,12].
The above statement is not entirely accurate. Great Britain (not the United Kingdom) has a mandatory revalidation system which includes CPD with the completion of four reflective cycles.
Lines 125 & 126
Data collection – pharmacists deemed as eligible were contacted. How did this happen? Emailed with the study information and consent form? Where did the list of eligible pharmacists come from? Were they identified by human resources dept within the hospitals?
Lines 152 & 153
To ensure that the guide was understandable and relevant the guide was piloted prior to the execution of the study.
With whom was this piloted?
Lines 155
…it was enhanced through recording and interpreting the verbal and non-verbal data.
You need to explain this as it was only audio recorded how was non-verbal data analysed? There is no mention of the analysis at this level in the results section.
Results section
Normally a minor theme is reported as a subtheme in qualitative work.
Are the 12 participants as representative sample of the hospital pharmacist population?
Lines 210 & 211
Repetition “Overall, the individuals interviewed recognized the importance of CPD in staying updated with industry developments, enhancing skills, and meeting regulatory requirements.” Already stated in 194/195

Some minor adjustments needed e.g line 103 should have the word respectively rather than receptively.
Highlighted some areas that need to be amended in the attached file.
Author Response
Reviewer 1
Brief Summary
- This paper aims to and provides some insights into how hospital pharmacists in SA view CPD and suggests some possible solutions to issues. A straightforward qualitative descriptive methodology was employed with the use of semi-structured interviews. Similar insights/results to previous studies were found. Some suggestions as to how the paper could be improved in relation to context and methodology details are outlined in more detail below.
Thank you very much for your encouraging remarks.
General Concept Comments
- The terms CE and CPD are sometimes used interchangeably in the literature. The FIP definition of CPD is included in the introduction which is helpful. CE systems typically collect points or hours of learning. CPD systems employed by pharmacy regulators have moved to more goal orientated learning with the use of reflective cycles. Other countries (e.g. Great Britain) have moved further again and introduced revalidations systems which incorporate peer discussions/reflections alongside CPD cycles. The CPD system in SA was mentioned but not described in terms of where it sits within this context. Does the pharmacy regulator in SA simply expect the pharmacists to submit certificates to provide evidence of hours at the end of each year? If this context was given in the introduction it would lead to more meaningful suggestions around the platform suggested in the discussion and conclusion. Would this platform require more goal orientated learning with the use of reflection and application of learning?
Thank you for bringing attention to this crucial point, which will undoubtedly help readers gain a clearer understanding of how CPD systems function in KSA. We have incorporated additional information into the text, highlighted in green for your reference.
- What postgraduate support is in place for qualified pharmacists? Is there a deanery/postgraduate provider or are universities offering courses? Or is it all CME which would normally be geared to medics?
At the postgraduate level, universities do not offer mandatory CPD/CME programs. However, many universities actively organize conferences and seminars to help clinicians earn CPD points and stay updated with advancements in their field.
Specific Comments
- Lines 59 and 60.
Today, many countries like the United States and British, have applied the mandatory set number of credits system for pharmacist license renewal [9,10]. Mandatory CPD is also used in the United Kingdom and Australia [11,12]. The above statement is not entirely accurate. Great Britain (not the United Kingdom) has a mandatory revalidation system which includes CPD with the completion of four reflective cycles.
Thanks, we have added the required information highlighted in green.
- Lines 125 & 126
Data collection – pharmacists deemed as eligible were contacted. How did this happen? Emailed with the study information and consent form? Where did the list of eligible pharmacists come from? Were they identified by human resources dept within the hospitals?
Thank you for the comment. More clarification about participant recruitment have been added in the text.
- Lines 152 & 153
To ensure that the guide was understandable and relevant the guide was piloted prior to the execution of the study.
With whom was this piloted?
Thank you very much for the comment. To ensure that the guide was understandable and relevant, the guide was reviewed by clinical pharmacists in the field. Then by academic experts in pharmacy practice department. More clarification has been added in the revised manuscript.
- Lines 155
…it was enhanced through recording and interpreting the verbal and non-verbal data.
You need to explain this as it was only audio recorded how was non-verbal data analysed? There is no mention of the analysis at this level in the results section.
Thank you for the comment. Non-verbal data, such as eye contact, facial expressions, gestures, and tone of voice, often provide valuable insights that enhance the understanding of verbal interviews. These cues can reveal emotions, attitudes, and reactions that may not be explicitly stated, helping researchers extract deeper meaning from participants' responses. Non-verbal data can be easily observed by analysing interview recordings, paying attention to elements like tone, pauses, and body language, which add richness and context to the verbal content. This helps in drawing more nuanced and accurate conclusions in qualitative research. We believe that nonverbal analysis is an integral part of verbal data analysis and doesn’t need more clarification.
Results Section
- Normally a minor theme is reported as a subtheme in qualitative work.
Thank you for the comment. Edited, as suggested (Highlighted in yellow).
- Are the 12 participants as representative sample of the hospital pharmacist population?
Thank you for the comment. Since the goal of qualitative research is to explore phenomena rather than to generalize findings to a broader population, we believe the sample size for this investigation was sufficient to effectively address the research objectives. The selected number of participants allowed for in-depth exploration and provided meaningful insights into the topic, aligning with the study's exploratory nature.
- Lines 210 & 211
Repetition “Overall, the individuals interviewed recognized the importance of CPD in staying updated with industry developments, enhancing skills, and meeting regulatory requirements.” Already stated in 194/195
Thank you for the comment. Edited, as suggested.

Reviewer 2 Report
Comments and Suggestions for Authors
Thank for your the opportunity to review this manuscript- it is an interesting study on an important topic. I offer the following comments for your consideration in revising the manuscript:
ABSTRACT: Please clarify that, even though the focus is on hospital pharmacists here, all pharmacists required to complete CPD.
INTRODUCTION:
* You seem to change between CE and CPD partway through the interaction and these are sometimes used interchangeably (which you know is not necessarily the case). The terms also have different meanings in different contexts. Please explain the difference between the two terms and how you understand/are using these here. If you are using them interchangeably, please pick one term and use it consistently.
* Your introduction could be more succinct, e.g. you don’t need examples of what counts as CPD
METHODS:
* I think you need a clearer rationale for focussing on hospital pharmacists at the end of your introduction/start of your methods section. This could just be reiterating or referring the reader back to the information in the introduction about the size and importance of the hospital pharmacy sector in Saudi Arabia, but further information is needed when you outline the rationale for your study.
* Please provide the abbreviation- COREQ- for the consolidated criteria for reporting of qualitative research
* Please describe process of thematic analysis (i.e. what you actually did when analysing this data) e.g. following Braun and Clarke, the following process for thematic analysis was used. 1 Broad reading of data to identify preliminary themes...
* When did data collection and analysis stop? I assume when thematic saturation was reached
DISCUSSION:
* Are there any limitations of the qualitative approach that should be mentioned here (e.g. lack of generalisability due not just to the setting but also the methods used)?
I look forward to reading the next version of the paper!
Comments on the Quality of English LanguageEnglish language was generally good. However, there were some minor spelling, grammatical and typographical errors e.g.
* Page 2 line 49: Should this be according to the latest statistics page 2 line 49 instead of the latest statistics;
* Page 4 line 127-128: Should it be For those who demonstrated interest….
* Page 4 line 129 should it be To impart credibility (not imparts)
Please double check revised paper for any further errors.
Author Response
Reviewer 2
- Thank for the opportunity to review this manuscript, it is an interesting study on an important topic. I offer the following comments for your consideration in revising the manuscript:
Thank you very much we appreciate your supportive comments.
ABSTRACT
- Please clarify that, even though the focus is on hospital pharmacists here, all pharmacists required to complete CPD.
Thank you for the comment. Added, as suggested (Highlighted in yellow).
INTRODUCTION
- You seem to change between CE and CPD partway through the interaction and these are sometimes used interchangeably (which you know is not necessarily the case). The terms also have different meanings in different contexts. Please explain the difference between the two terms and how you understand/are using these here. If you are using them interchangeably, please pick one term and use it consistently.
Thank you very much for the comment, we have changed it to the CPD throughout the manuscript now. Additionally, following text has been added to differentiate between CPD and CE/CME.
“In Saudi Arabia, Saudi Commission of Health Specialties (SCFHS) oversees CPD activities which includes health and medical training courses, conferences, seminars, or lectures designed to advance health practitioners' skills and to obtain CME (Continuous Medical Education) hours…”
- Your introduction could be more succinct, e.g. you don’t need examples of what counts as CPD.
Thank you for the suggestion. However, we believe it is important to include examples of CPD activities, as the types of activities considered part of CPD can vary by region. Since the introduction already discusses various CPD practices from around the world, providing specific examples will offer readers clearer context and understanding. We hope this approach is acceptable to you.
METHODS
- I think you need a clearer rationale for focussing on hospital pharmacists at the end of your introduction/start of your methods section. This could just be reiterating or referring the reader back to the information in the introduction about the size and importance of the hospital pharmacy sector in Saudi Arabia, but further information is needed when you outline the rationale for your study.
Many Thanks, added as suggested (Highlighted in yellow).
- Please provide the abbreviation- COREQ- for the consolidated criteria for reporting of qualitative research
Many Thanks, added as suggested (Highlighted in yellow).
- Please describe process of thematic analysis (i.e. what you actually did when analysing this data) e.g. following Braun and Clarke, the following process for thematic analysis was used. 1 Broad reading of data to identify preliminary themes...
Many Thanks, added as suggested (Highlighted in yellow).
- When did data collection and analysis stop? I assume when thematic saturation was reached
Many Thanks, it has been added (Highlighted in yellow).
DISCUSSION
- Are there any limitations of the qualitative approach that should be mentioned here (e.g. lack of generalisability due not just to the setting but also the methods used)?
Thanks, it has been added.
- I look forward to reading the next version of the paper!
Thank you, we much appreciate your valuable comments and hope that we addressed all these comments
Comments on the Quality of English Language
- English language was generally good. However, there were some minor spelling, grammatical and typographical errors e.g.
- Page 2 line 49: Should this be according to thelatest statistics page 2 line 49 instead of the latest statistics;
- Page 4 line 127-128: Should it be Forthose who demonstrated interest….
- Page 4 line 129 should it be Toimpart credibility (not imparts)
- Please double check revised paper for any further errors.
Thank you for your time and efforts to give us these corrections, I have reviewed and corrected all these notes (Highlighted in yellow). Thanks

Round 2
Reviewer 2 Report
Comments and Suggestions for Authors
Thank you for your efforts revising this paper. All of my comments have been addressed and it is now suitable for publication in its current form.